

# Simulating highly disturbed vegetation distribution: the case of China's Jing-Jin-Ji region

Sangui Yi[1,2], Jihua Zhou[1], Liming Lai[1], Hui Du[1], Qinglin Sun[1,2], Liu Yang[1,2], Xin Liu[1,2], Benben Liu[1,2] and Yuanrun Zheng[1]

[1] Key Laboratory of Plant Resources, West China Subalpine Botanical Garden, Institute of Botany, Chinese Academy of Sciences, Beijing, China
[2] University of Chinese Academy of Sciences, Beijing, China

Corresponding author
Yuanrun Zheng, zhengyr@ibcas.ac.cn

## ABSTRACT

**Background**. Simulating vegetation distribution is an effective method for identifying vegetation distribution patterns and trends. The primary goal of this study was to determine the best simulation method for a vegetation in an area that is heavily affected by human disturbance.

**Methods**. We used climate, topographic, and spectral data as the input variables for four machine learning models (random forest (RF), decision tree (DT), support vector machine (SVM), and maximum likelihood classification (MLC)) on three vegetation classification units (vegetation group (I), vegetation type (II), and formation and subformation (III)) in Jing-Jin-Ji, one of China's most developed regions. We used a total of 2,789 vegetation points for model training and 974 vegetation points for model assessment.

**Results**. Our results showed that the RF method was the best of the four models, as it could effectively simulate vegetation distribution in all three classification units. The DT method could only simulate vegetation distribution in units I and II, while the other two models could not simulate vegetation distribution in any of the units. Kappa coefficients indicated that the DT and RF methods had more accurate predictions for units I and II than for unit III. The three vegetation classification units were most affected by six variables: three climate variables (annual mean temperature, mean diurnal range, and annual precipitation), one geospatial variable (slope), and two spectral variables (Mid-infrared ratio of winter vegetation index and brightness index of summer vegetation index). Variables Combination 7, including annual mean temperature, annual precipitation, mean diurnal range and precipitation of driest month, produced the highest simulation accuracy.

**Conclusions**. We determined that the RF model was the most effective for simulating vegetation distribution in all classification units present in the Jing-Jin-Ji region. The RF model produced high accuracy vegetation distributions in classification units I and II, but relatively low accuracy in classification unit III. Four climate variables were sufficient for vegetation distribution simulation in such region.

## INTRODUCTION

Vegetation is an essential component of terrestrial ecosystems and landscapes (*Editorial Committee of Vegetation Map of China, 2007*). Environmental research, resource management, and conservation planning require vegetation distribution maps (*Franklin, 2010*) to better understand, use, and monitor vegetation. Vegetation patterns and distributions are affected by the climate (*Chen et al., 2015*; *Zhang et al., 2018*) and other disturbances, particularly those caused by changes in land use (*Hansen et al., 2013*; *Wehkamp et al., 2018*). Human disturbances, such as industrialization, urbanization, population growth, land use change for agricultural use, etc., strongly influence the environment by greatly altering vegetation patterns, making exact mapping a significant challenge (*Xie, Sha & Yu, 2008*; *Zhou et al., 2016*).

Field surveys, the traditional method used to map vegetation, are costly and labor-intensive (*Newell & Leathwick, 2005*; *Zhou et al., 2016*). Mapping using remote sensing data is also a popular method that has been used over the last 30 years (*Xie, Sha & Yu, 2008*). This method makes it possible to obtain a wide range of reliable data from remote sensing images, and it updates vegetation boundaries by visually interpreting images and field surveys (*Zhang et al., 2008*). However, determining vegetation units and their boundaries by visual interpretation can produce inaccurate results. Researchers may get different results from the same images for the same study areas (*Bie & Beckett, 1973*; *Pfeffer, Pebesma & Burrough, 2003*). Furthermore, field survey and remote sensing methods manually draw vegetation unit boundaries based on climate, elevation, and soil type information, which can be inaccurate in transition areas (*Zhang et al., 2008*). Using simulation models in combination with field and remote sensing data may be an effective alternative for mapping vegetation.

Changes in the environment can affect vegetation composition, structure, function, and spatial distribution. Environmental variables have been used to simulate the global distribution of vegetation (*Dilts et al., 2015*; *Mod et al., 2016*). Simulation models are usually developed to test how environmental variables control vegetation distribution (*Guisan & Zimmermann, 2000*). Modern remote sensing data and software make it more convenient than ever before to produce predictive vegetation maps (*Franklin, 1995*).

Predictive vegetation mapping uses environmental variables and various models based on niche theory and gradient analysis to visualize communities in geographic space (*Dilts et al., 2015*; *Lany et al., 2019*). Other methods based on statistics and machine learning have also been used to simulate vegetation distribution. Predictive vegetation mapping includes various statistical methods such as the generalized linear model, the generalized additive model, and multivariate statistical approaches (*Lany et al., 2019*; *Prasad, Iverson & Liaw, 2006*). Recently, machine learning modeling methods have been used to map the distribution of both vegetation communities and individual species. These methods include the support vector machine (SVM), decision tree (DT), and artificial neural network (*Guisan & Zimmermann, 2000*; *Hastie, Tibshirani & Friedman, 2009*; *Zhou et al., 2016*). These machine learning models have fewer limitations and can produce more reliable results than traditional vegetation modeling methods (*Hastie, Tibshirani & Friedman,*

*2009*). Advanced machine learning techniques can integrate spectral and spatial predictors and improve classification accuracy by retaining important information about vegetation composition and structural differences (*Sesnie et al., 2010*). Machine learning models efficiently and cost-effectively produce vegetation maps without the general inaccuracies caused by visual interpretation (*Franklin, 2010*).

The Jing-Jin-Ji region, also known as the Beijing-Tianjin-Hebei urban agglomeration, is the center of northern Chinese politics, culture, and economy. Because of its extension, it faces significant problems such as unbalanced regional development and the struggle between economic growth and limited resources. The region's larger cities, including Beijing and Tianjin, have large populations, developed economies, and abundant educational resources. However, these big cities face issues of limited natural resources and serious ecological and environmental pollution. In particular, Beijing's large population requires limited resources such as water, land, and vegetation (*Wang & Gong, 2018*). Breaking up administrative divisions may be the best method to coordinate regional development (*Wang et al., 2019*). The new Xiong'an area located in Hebei province is being constructed to relocate some of Beijing's population. The development of areas like Xiong'an is affected by the surrounding natural environment. To better integrate the environmental carrying capacity and socioeconomic development of the Jing-Jin-Ji region, including the new Xiong'an area, accurate vegetation maps with temporal resolution are necessary. The most updated vegetation map of the Jing-Jin-Ji region is the Vegetation Map of the People's Republic of China (VMC), with a scale of 1:1,000,000 (*Editorial Committee of Vegetation Map of China, 2007*). Most of its data come from a field survey conducted between 1980 and 1990, meaning its temporal and spatial scales are both outdated.

In this study, we integrated geospatial, climate, and spectral data to simulate vegetation distribution through four different models across three vegetation classification units. This research was different from the research of *Zhou et al. (2016)*. Firstly, the research area of this research was the Jing-Jin-Ji region located in the North China Plain and affected by high social-economic disturbance, while the Qilian Mountain in the research of Zhou et al. is characterized by complex terrain, but without high social-economic disturbance. Secondly, the predictive variables as well as the combinations of these variables were different from the research of *Zhou et al. (2016)*. Thirdly, we compared four model methods for simulating distribution of vegetation in three vegetation classification levels, while only three models were used for simulation in two vegetation classification levels in the research of *Zhou et al. (2016)*. Our primary objectives were to: (1) determine the best modeling method for vegetation affected by high socioeconomic disturbance, (2) create an improved vegetation map of the Jing-Jin-Ji region, (3) determine the predictive abilities of different models across different vegetation classification units, and (4) determine which variables enhanced the classification accuracy for vegetation mapping.

## MATERIALS & METHODS

### Study area

The Jing-Jin-Ji region is located in the northern part of the North China Plain. Its location ranges from 113°04′ to 119°53′E and 36°01′ to 42°37′N and is bordered by Taihang

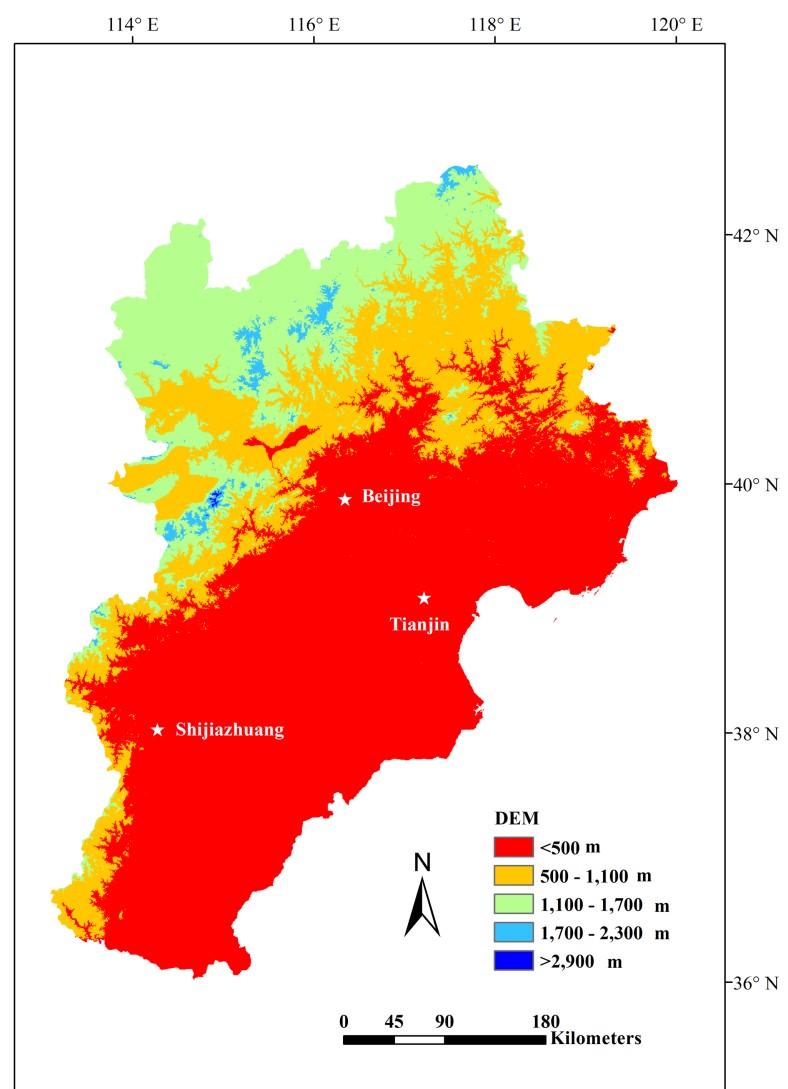

**Figure 1** **The location and DEM of the Jing-Jin-Ji region.**

Mountain in the west, Yanshan Mountain in the north, and the Bohai Sea in the east. The region includes the Beijing, Tianjin, and Hebei provinces (Fig. 1). Jing-Jin-Ji has a population of approximately 110 million people and covers an area of approximately 216,000 km$^2$ (*Wang et al., 2019*). The region is a temperate monsoon climate zone with an elevation range of −14 to 2,837 m (Fig. 1). The annual precipitation ranges from 305 to 711 mm, with increased precipitation at lower altitudes. The annual mean temperature ranges from −3 to 14 °C, with colder averages at higher elevations. The amount of precipitation in the region gradually decreases moving from the southeast to the northwest, while temperature changes show the reverse pattern.

## Vegetation and training data

The VMC, completed in 2007 based on field survey data, included eight vegetation groups (I), 15 vegetation types (II), and 75 formations and subformations (III) from the Jing-Jin-Ji region. However, some of the map's vegetation unit areas are very small and difficult to distinguish. To ensure that enough training and assessment point data can be randomly selected in units II and III, we selected eight units I, 12 units II, and 39 units III from the study area (Table 1). Cultivated vegetation are mainly distributed in low areas with an altitude range of −14 to 254 m and an annual mean temperature range of 7 to 14 °C. Major cultivated plants include winter wheat and coarse grains. Scrub and grass-forb communities are mainly distributed in the north, in elevations ranging from 254 to 1,440 m.

We obtained model training and assessment data on vegetation composition from field surveys and other publications. We collected a total of 3,763 vegetation points, with 2,789 of those used for model training and 974 used for model assessment. Each unit III had at least 80 vegetation points, with at least 60 of those used for model training and 20 used for model assessment. The model training and assessment data were randomly selected for each unit III. Additionally, we increased the credibility of the model assessment by first rasterizing the vector VMC onto the same grid as the modeled data, and then assessing the data using the Kappa coefficient (*Landis & Koch, 1977*; *Weng & Zhou, 2006*; *Zhou et al., 2016*).

## Geospatial, climate, and spectral data

We derived geospatial variables, including elevation, slope, and aspect, from the 30 m resolution Shuttle Radar Topography Mission (SRTM) digital elevation model (DEM; *Zhao et al., 2018*). We then resampled these data to a 500×500 m grid cell size using the cubic technique in ArcGIS 10.3 (*Wu et al., 2019*).

We downloaded the climate data, including 19 bioclimatic variables, at ∼1 km resolution from WorldClim (*Fick & Hijmans, 2017*) at http://worldclim.org/. These climate data were also resampled to a 500 × 500 m grid cell size using the cubic technique in ArcGIS 10.3 (*Wu et al., 2019*). Climatic variables are important for plant ecophysiology (*Mod et al., 2016*) and are commonly used as bioclimatic limits in vegetation models (*Sitch et al., 2003*).

We acquired the MYD09A1500M product data (sinusoidal projection, path 4 and row 26, path 4 and row 27, path 5 and row 26, path 5 and row 27) from summer (July 20, 2013) and winter (January 17, 2013) as Modis images from the Geospatial Data Cloud at http://www.gscloud.cn/. Our image pre-processing included image subset mosaicking and clipping in ENVI 5.2 (*Deng, 2010*). We obtained the land surface albedo in bands 1-7 directly from the MYD09A1500M product, and calculated the indices' effectiveness at reflecting vegetation information (*Price, Guo & Stiles, 2002*; *Zhou et al., 2016*).

Since vegetation indices can provide information on both vegetation and environment (*Bannari, Morin & Bonn, 1995*), these indices are more sensitive than single spectral bands at detecting green vegetation (*Bannari, Morin & Bonn, 1995*; *Cohen & Goward, 2004*). Therefore, vegetation indices can be used for image interpretation, vegetation discrimination and prediction, and land use change detection (*Bannari, Morin & Bonn,*
**Table 1** Classification units of the vegetation of China.

| Vegetation groups (I) | Vegetation types (II) | Formations and sub-formations (III) |
|---|---|---|
| 0. No vegetation | 0 No vegetation | 0 No vegetation |
| 1. Needleleaf forest | 1 Temperate needleleaf forest | 1 *Pinus tabulaeformis* forest |
| 2. Broadleaf forest | 2 Temperate broadleaf deciduous forest | 2 *Quercus mongolica* forest |
| | | 3 *Quercus liaotungensis* forest |
| | | 4 *Quercus variabilis* forest |
| | | 5 *Robinia pseudoacacia* forest |
| | | 6 *Salix matsudana* forest |
| | | 7 *Populus davidiana* forest |
| | | 8 *Betula platyphylla* forest |
| 3. Scrub | 3 Temperate broadleaf deciduous scrub | 9 *Corylus heterophylla* scrub |
| | | 10 *Lespedeza bicolor* scrub |
| | | 11 *Prunus armeniaca var. ansa* scrub |
| | | 12 *Vitex negundo* var. *heterophylla*, *Zizyphus jujuba* var. *spinosa* scrub |
| | | 13 *Cotinus coggygria* var. *cinerea* scrub |
| | | 14 *Spiraea spp.* scrub |
| | | 15 *Ostryopsis davidiana* scrub |
| 4. Steppe | 4 Temperate grass-forb meadow steppe | 16 *Stipa baicalensis,* forb meadow steppe |
| | | 17 *Filifolium sibiricum,* grass-forb meadow steppe |
| | 5 Temperate needlegrass arid steppe | 18 *Aneurolepidium chinense*, needlegrass steppe |
| | | 19 *Stipa krylovii* steppe |
| | | 20 *Stipa bungiana* steppe |
| | | 21 *Thymus mongolicus,* needlegrass steppe |
| 5. Grass-forb community | 6 Temperate grass-forb community | 22 *Bothriochloa ischaemum* community |
| | | 23 *Bothriochloa ischaemum* community |
| | | 24 *Vitex negundo* var. *heterophylla*, *Zizyphus jujuba* var. *spinosus*, *Bothriochloa ischaemum* scrub and grass community |
| | | 25 *Vitex negundo* var. *heterophylla*, *Zizyphus jujuba* var. *spinosus*, *Themeda triandra* var. *japonica* scrub and grass community |
| 6. Meadow | 7 Temperate grass and forb meadow | 26 *Arundinella hirta, Spodiopogon sibiricus*, forb meadow |
| | | 27 *Carex spp.,* forb meadow |
| | 8 Temperate grass and forb holophytic meadow | 28 *Achnatherum splendens* holophytic meadow |
| | | 29 *Suaeda glauca* holophytic meadow |
| 7. Swamp | 9 Cold-temperate and temperate swamp | 30 *Phragmites communis* swamp |

**Table 1** (*continued*)

| Vegetation groups (I) | Vegetation types (II) | Formations and sub-formations (III) |
|---|---|---|
| | 10 One crop annually and cold-resistant economic crops | 31 Spring wheat, naked oats, buckwheat, potatoes; flux |
| | 11 One crop annually, cold-resistant economic crops and deciduous orchards | 32 Coarse grains |
| | | 33 Winter wheat, coarse grains |
| | | 34 Coarse grains |
| | | 35 Rice |
| 8. Cultural vegetation | | 36 Winter wheat, corn, cotton |
| | 12 Three crops two years and two crops annually non irrigation, deciduous orchards | 37 Apple, pear orchard |
| | | 38 Winter wheat, corn, Chinese sorghum, sweet potatoes; cotton, tabacco, peanut, sesame; apple, pear, hauthorn, persimmon, walnut, pomegranat, grape |
| | | 39 Winter wheat, coarse grains (loamy soil) |

**Table 2 The vegetation indices.**

| Indices | Abbreviation | Formula |
|---|---|---|
| Ratio vegetation index | RVI | NIR/Red |
| Brightness index | BI | 0.2909Blue + 0.2493Green + 0.4806Red + 0.5568NIR + 0.4438SWIR1 + 0.1706SWIR2 |
| Green vegetation index | GI | −0.2728Blue - 0.2174Green-0.5508Red + 0.7221NIR + 0.0733SWIR1 - 0.1648SWIR2 |
| Wetness index | WI | 0.1446Blue + 0.1761Green + 0.3322Red + 0.3396NIR - 0.6210SWIR1 - 0.4186SWIR2 |
| Differenced vegetation index | DVI | NIR - Red |
| Green ratio | GR | NIR/Green |
| Mid-infrared ratio | MR | NIR/SWIR1 |
| Soil-adjusted vegetation index | SAVI | (1.5(NIR - Red))/((NIR + Red + 0.5)) |
| Optimization of soil-adjusted vegetation index | OSAVI | (1.16(NIR - Red))/((NIR + Red + 0.16)) |
| Atmospherically resistant vegetation index | ARVI | (NIR - (2*Red - Blue))/(NIR + (2*Red - Blue)) |
| Normalized difference vegetation index | NDVI | (NIR - Red)/(NIR + Red) |
| Enhanced vegetation index | EVI | 2.5[(NIR - Red)/(NIR + 6*Red - 7.5Blue + 1)] |
| Normalized difference tillage index | NDTI | (SWIR1-SWIR2)/(SWIR1 + SWIR2) |
| Normalized difference senescent vegetation index | NDSVI | (SWIR1-Red)/(SWIR1 + Red) |

*1995*; *Cohen & Goward, 2004*; *Zhou et al., 2016*). We tested the vegetation discrimination of 14 vegetation indices (Table 2).

To determine the distribution predictive ability of different variables, we grouped the variables into different combinations based on the results of the Pearson correlation. We only used less correlated variables (R < |0.7|, Pearson correlation) (*Chala et al., 2017*) in Combinations 1–9 (Table 3), then used variable combinations as input predictor variables to simulate vegetation distribution. Combination 1 included the less correlated variables of the summer land surface albedos from bands 1 to 7. Combination 2 included the less correlated variables of the winter land surface albedos from bands 1 to 7. Combination 3 included the less correlated variables in Combinations 1 and 2. Combination 4 included

**Table 3  Variable combinations.** Note: DT10 and RF10 represent the top 10 important variables of decision tree (DT) and random forest (RF) methods with Combination 9 in the vegetation group level, respectively. The vegetation indices and their abbreviations were shown in Table 2.

| Number | Variables combinations |
| --- | --- |
| 1 | Summer land surface albedos of band 1 and 5. |
| 2 | Winter land surface albedos of band 1 and 6. |
| 3 | Summer land surface albedos of band 1 and 5. Winter land surface albedos of band 1 and 6. |
| 4 | Summer vegetation indices BI, WI, MR, NDVI, EVI. |
| 5 | Winter vegetation indices MR, NDVI, EVI, NDTI. |
| 6 | Summer vegetation indices BI, WI, MR, NDVI, EVI. Winter vegetation indices MR, NDVI, EVI, NDTI. |
| 7 | Annual mean temperature, Annual precipitation, Mean diurnal range, Precipitation of driest month. |
| 8 | Slope, Aspect, Annual mean temperature, Annual precipitation, Mean diurnal range, Precipitation of driest month. |
| 9 | Summer land surface albedos of band 1. Winter land surface albedos of band 6. Summer vegetation indices BI, WI, MR, NDVI, EVI. Winter vegetation indices MR, NDVI, EVI, NDTI. Slope, Aspect, Annual mean temperature, Annual precipitation, Mean diurnal range, Precipitation of driest month. |
| 10 | DT10: Annual mean temperature, Annual precipitation, Mean diurnal range, Precipitation of driest month, Slope, Winter vegetation indices MR, Summer land surface albedos of band 1, Summer vegetation indices BI and EVI, Winter land surface albedos of band 6. |
| 11 | RF10: Annual precipitation, Annual mean temperature, Mean diurnal range, Slope, Summer vegetation indices BI, MR, NDVI and EVI, Winter vegetation indices MR and NDVI. |

the less correlated variables of the summer vegetation indices. Combination 5 included the less correlated variables of the winter vegetation indices. Combination 6 included the less correlated variables in Combinations 4 and 5. Combination 7 included the less correlated variables from the 19 bioclimatic variables. Combination 8 included the less correlated variables from the 19 bioclimatic variables and three geospatial variables. Combination 9 included the less correlated variables in Combinations 3, 6, and 8. Combinations 10 and 11 represented the top 10 most important variables of the DT and RF methods, with Combination 9 in vegetation unit I, respectively (Table 3). The SVM and maximum likelihood classification (MLC) methods only output the simulation results of variable Combinations 1 to 6, likely due to the training samples' weak separability (*Deng, 2010*).

## Vegetation distribution models

We used DT, RF, MLC, and SVM vegetation distribution models in this study. The DT model is a divisive, monothetic, and supervised classifier often used for species distribution modeling and related applications (*Franklin, 2010*). It is computationally fast

and easy to understand and implement. It uses classification or regression algorithms to generate classification rules, and then visualizes those rules into simple tree graphics (*Hastie, Tibshirani & Friedman, 2009*; *Zhou et al., 2016*). The DT model calculates the most significant variables contributing to the model (*Deng, 2010*). We used a DT with five layers, 40 samples in the smallest parent node, and 10 samples in the smallest child node.

The RF model is an ensemble method that has been applied in risk assessment and species distribution modeling studies (*Cutler et al., 2007*; *Zhang & Dong, 2017*). The RF model creates and combines different DTs to produce considerably more accurate classifications that are unaffected by noise or overtraining (*Burai et al., 2015*; *Cutler et al., 2007*; *Gislason, Benediktsson & Sveinsson, 2006*). The RF model also calculates the most significant variables that contribute to the model (*Cutler et al., 2007*). Running an RF model requires defined parameters, including tree number, number of randomly selected features, and node impurity function. We generated the RF model in EnMAP-Box, a license-free and platform-independent software interface designed to process hyperspectral remote sensing data, which was developed by the Humboldt University of Berlin. There are in-built applications aimed at the processing of hyperspectral data, such as SVM and RF for classification of image data in the EnMAP-Box (*Held et al., 2014*). We used the default settings in EnMAP-Box with 100 trees. The number of randomly selected features was equal to the square root of the number of all features, and we used a Gini coefficient for the node impurity function (*Rabe et al., 2014*; *Ma, Gao & Gu, 2019*; *van der Linden et al., 2015*; *Zhou et al., 2016*).

The MLC model is one of the most commonly used supervised image classification methods. MLC's classification rules use the statistics of the Gaussian probability density function to assign each pixel to the class with the highest probability. Although the MLC method usually generates similar or more accurate classifications than other methods, it is not applicable when there are fewer training samples than input predictors (*Burai et al., 2015*; *Zhou et al., 2016*).

The SVM model is a supervised machine learning model used for classification and regression. It is a complex and widely used method that can output more accurate predictions (*Burai et al., 2015*) than other methods. The SVM model searches for an optimal plane in a multidimensional space to divide all sample elements into two categories, making the distance between the closest points in the two classes as large as possible (*Kabacoff, 2016*). Running an SVM model requires a defined kernel parameter g and regularization parameter c. In this study, we generated the SVM model in the EnMAP-Box. The default settings in EnMAP-Box to the SVM model was applied, where the parameter g was 0.01 to 1,000, and the parameter c was 0.1 to 1,000. Parameters g and c were tested using a grid search with internal performance estimation, and we used those with the best performance for data training (*Lin et al., 2014*; *van der Linden et al., 2014*; *van der Linden et al., 2015*).

We generated the predicted vegetation maps of the three classification units using the DT, RF, MLC, and SVM methods with a resolution of 500 m. We selected all 11 variable combinations as the input variables for each method. The DT and RF method results indicated which variables were most important for vegetation discrimination.

## Model assessment

We used the VMC and a total of 974 vegetation points to assess the overall accuracy and Kappa coefficient of every predicted vegetation map. Kappa coefficient values ranging from 0.4 to 0.55 indicated moderate agreement, from 0.56 to 0.8 indicated substantial agreement, and from 0.81 to 1 indicated almost perfect agreement (*Landis & Koch, 1977*; *Weng & Zhou, 2006*; *Zhou et al., 2016*). When the Kappa coefficient value was greater than 0.4, the assessed predicted map was considered acceptable.

## RESULTS

### Unit I modeling and assessment

The RF model's results were better than the results of the DT, MLC, and SVM models (Table 4). The RF model had a Kappa coefficient larger than 0.4 when using variable Combinations 6 to 11 assessed by field point data and VMC assessments, respectively. The RF model had a Kappa coefficient larger than 0.4 when using variable Combinations 7 to 11 assessed by field data, with an overall accuracy of 68% to 72%. The RF model had the highest Kappa coefficient of 0.66 and the highest overall accuracy of 72% when using variable Combination 7. The DT model had a Kappa coefficient larger than 0.4 when using variable Combinations 7 to 11 assessed by field point data, with an overall accuracy of 54% to 56%. The DT model had no Kappa coefficient larger than 0.56 when using all variable combinations. After VMC assessment, we found the highest Kappa coefficient was 0.38 with an overall accuracy of 57% in the RF model using variable Combinations 9 to 11 (Table 4; Fig. 2).

### Unit II modeling and assessment

The RF model results were better than the results of the other three models. The RF model using variable Combinations 7 to 11 had a Kappa coefficient larger than 0.4, with overall accuracies of 66%–70% and 54%–55% for field point data and VMC assessments, respectively. The RF model using Combinations 7 to 11 had a Kappa coefficient larger than 0.56 and an overall accuracy of 66%–70% when assessed by field point data. The RF model had the highest Kappa coefficient of 0.65 and the highest overall accuracy of 70% when using variable Combination 7. The DT model using variable Combinations 7 to 11 had a Kappa coefficient larger than 0.4, with overall accuracies of 53%–55% and 65%–72% for field point data and VMC assessments, respectively. The DT model had the highest Kappa coefficient of 0.54 and overall accuracy of 72% when using variable Combination 7. The DT model had a larger Kappa coefficient and greater overall accuracy when assessed by VMC rather than the RF model (Table 5; Fig. 3).

### Unit III modeling and assessment

Only the RF model could simulate vegetation distribution in unit III. The RF model using variable Combinations 7 to 11 had a Kappa coefficient larger than 0.4 and an overall accuracy of 55%–58% assessed by field point data. The RF model using variable Combination 7 had the highest Kappa coefficient of 0.57 (the only model with a Kappa coefficient larger than 0.56) and the highest overall accuracy of 58% assessed by field point

**Table 4  Model assessment of vegetation groups by field point data and VMC.** Variable combinations were shown in Table 3.

| Variable combinations | Decision tree | | | | Random forest | | | | Support vector machine | | | | Maximum likelihood classification | | | |
|---|---|---|---|---|---|---|---|---|---|---|---|---|---|---|---|---|
| | Point data | | VMC | | Point data | | VMC | | Point data | | VMC | | Point data | | VMC | |
| | OA | KC | OA | KC | OA | KC | OA | KC | OA | KC | OA | KC | OA | KC | OA | KC |
| 1 | 34% | 0.18 | 55% | 0.22 | 37% | 0.24 | 32% | 0.09 | 36% | 0.21 | 53% | 0.21 | 23% | 0.08 | 11% | 0.02 |
| 2 | 38% | 0.20 | 52% | 0.23 | 39% | 0.27 | 37% | 0.13 | 35% | 0.20 | 55% | 0.24 | 18% | 0.07 | 9% | 0.03 |
| 3 | 45% | 0.31 | 54% | 0.26 | 47% | 0.36 | 45% | 0.21 | 41% | 0.27 | 54% | 0.27 | 24% | 0.12 | 15% | 0.05 |
| 4 | 32% | 0.16 | 46% | 0.16 | 42% | 0.30 | 42% | 0.17 | 37% | 0.22 | 57% | 0.26 | 11% | 0.04 | 3% | 0.01 |
| 5 | 31% | 0.11 | 59% | 0.14 | 44% | 0.32 | 44% | 0.19 | 36% | 0.22 | 51% | 0.22 | 9% | 0.04 | 4% | 0.02 |
| 6 | 41% | 0.26 | 44% | 0.18 | 50% | 0.40[*] | 52% | 0.27 | 42% | 0.29 | 54% | 0.27 | 13% | 0.08 | 4% | 0.03 |
| 7 | 54% | 0.45[*] | 57% | 0.34 | 72% | 0.66[**] | 55% | 0.35 | | | | | | | | |
| 8 | 55% | 0.46[*] | 56% | 0.35 | 69% | 0.63[**] | 56% | 0.37 | | | | | | | | |
| 9 | 55% | 0.46[*] | 53% | 0.34 | 68% | 0.61[**] | 57% | 0.38 | | | | | | | | |
| 10 | 55% | 0.46[*] | 53% | 0.33 | 69% | 0.63[**] | 57% | 0.38 | | | | | | | | |
| 11 | 56% | 0.46[*] | 56% | 0.36 | 68% | 0.62[**] | 57% | 0.38 | | | | | | | | |

**Notes.**
VMC, the Vegetation Map of the Peoples Republic of China.
[**]the kappa coefficient lager than 0.56.
[*]the kappa coefficient larger than 0.4 and less than 0.56.
OA, Overall accuracy, KC, Kappa coefficient.

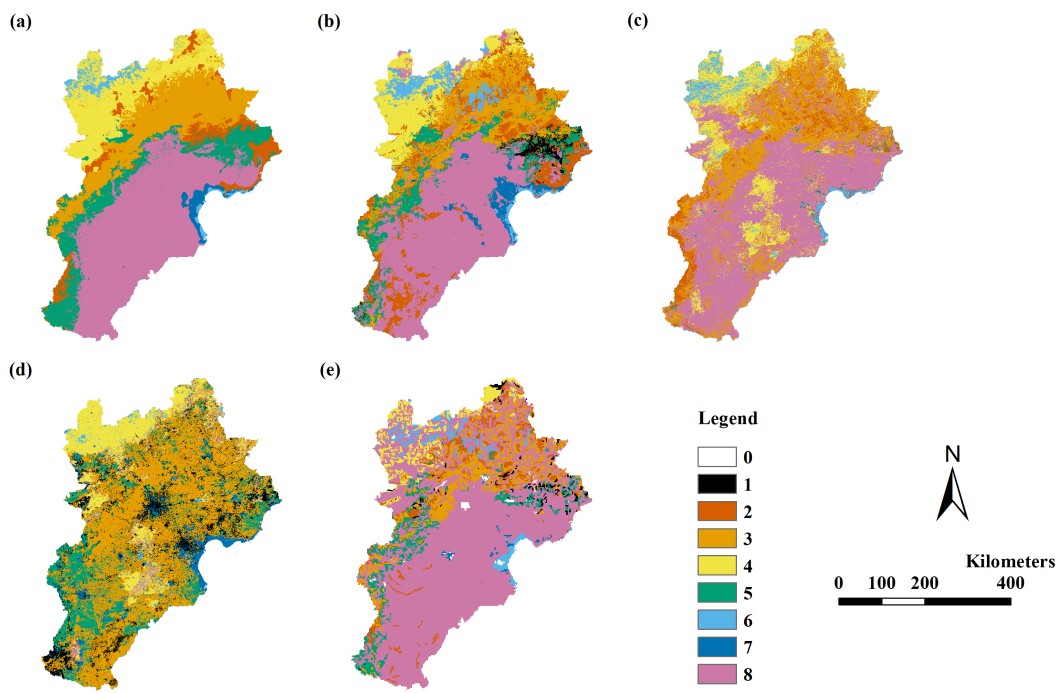

**Figure 2 The modeling vegetation map of vegetation groups with highest accuracy by four methods and the VMC in Jing-Jin-Ji region.** Decision tree model (A), random forest model (B), support vector machine (C), maximum likelihood classification (D), the Vegetation Map of the People's Republic of China (VMC) (E). The legend represents vegetation groups shown in Table 1.

data. The Kappa coefficients in all models were less than 0.4 when assessed by the VMC (Table 6; Fig. 4).

The abbreviations were same with Table 4.

## Important variables

For the RF model, eight of the top 10 most important variables were the same across the different vegetation units: three climate variables (annual mean temperature, mean diurnal range, and annual precipitation), one geospatial variable (slope), and four spectral variables (Mid-infrared ratio and NDVI of winter vegetation index, brightness index and NDVI of summer vegetation index). For the DT model, nine of the top 10 most important variables were the same across the different vegetation units: four climate variables (annual mean temperature, mean diurnal range, precipitation of the driest month, and annual precipitation), one geospatial variable (slope), and 4 spectral variables (Mid-infrared ratio of winter vegetation index, brightness index of summer vegetation index, summer surface albedo of band 1, winter surface albedo of band 6) (Table 7).

## DISCUSSION

### Vegetation classification units

Vegetation classification is an important and complex system with multiple levels. Higher level classification methods not only accurately classify vegetation, but they can also describe

Yi et al. (2020), *PeerJ*, DOI 10.7717/peerj.9839

**Table 5 Model assessment of vegetation types by field point data and VMC.** Variable combinations were shown in Table 3.

| Variable combinations | Decision tree | | | | Random forest | | | | Support vector machine | | | | Maximum likelihood classification | | | |
|---|---|---|---|---|---|---|---|---|---|---|---|---|---|---|---|---|
| | Point data | | VMC | | Point data | | VMC | | Point data | | VMC | | Point data | | VMC | |
| | OA | KC | OA | KC | OA | KC | OA | KC | OA | KC | OA | KC | OA | KC | OA | KC |
| 1 | 42% | 0.24 | 63% | 0.33 | 32% | 0.22 | 23% | 0.09 | 32% | 0.18 | 40% | 0.18 | 6% | 0.02 | 7% | 0.00 |
| 2 | 44% | 0.27 | 58% | 0.31 | 34% | 0.23 | 30% | 0.14 | 31% | 0.18 | 44% | 0.24 | 5% | 0.02 | 14% | 0.00 |
| 3 | 43% | 0.30 | 58% | 0.35 | 44% | 0.34 | 38% | 0.22 | 37% | 0.26 | 43% | 0.25 | 9% | 0.05 | 13% | 0.00 |
| 4 | 36% | 0.20 | 47% | 0.20 | 39% | 0.29 | 31% | 0.15 | 32% | 0.19 | 43% | 0.21 | 13% | 0.07 | 6% | 0.02 |
| 5 | 32% | 0.14 | 59% | 0.23 | 41% | 0.31 | 36% | 0.19 | 34% | 0.22 | 43% | 0.22 | 6% | 0.03 | 6% | 0.03 |
| 6 | 36% | 0.23 | 45% | 0.24 | 47% | 0.38 | 44% | 0.27 | 40% | 0.29 | 43% | 0.25 | 14% | 0.09 | 21% | 0.06 |
| 7 | 55% | 0.46[*] | 72% | 0.54[*] | 70% | 0.65[**] | 54% | 0.41[*] | | | | | | | | |
| 8 | 53% | 0.44[*] | 68% | 0.52[*] | 68% | 0.63[**] | 55% | 0.43[*] | | | | | | | | |
| 9 | 54% | 0.45[*] | 65% | 0.49[*] | 66% | 0.60[**] | 55% | 0.43[*] | | | | | | | | |
| 10 | 54% | 0.45[*] | 65% | 0.49[*] | 68% | 0.63[**] | 55% | 0.43[*] | | | | | | | | |
| 11 | 53% | 0.44[*] | 68% | 0.52[*] | 67% | 0.62[**] | 55% | 0.43[*] | | | | | | | | |

**Notes.**

VMC, the Vegetation Map of the Peoples Republic of China.

[**] the kappa coefficient lager than 0.56.

[*] the kappa coefficient larger than 0.4 and less than 0.56.

OA, Overall accuracy, KC, Kappa coefficient.

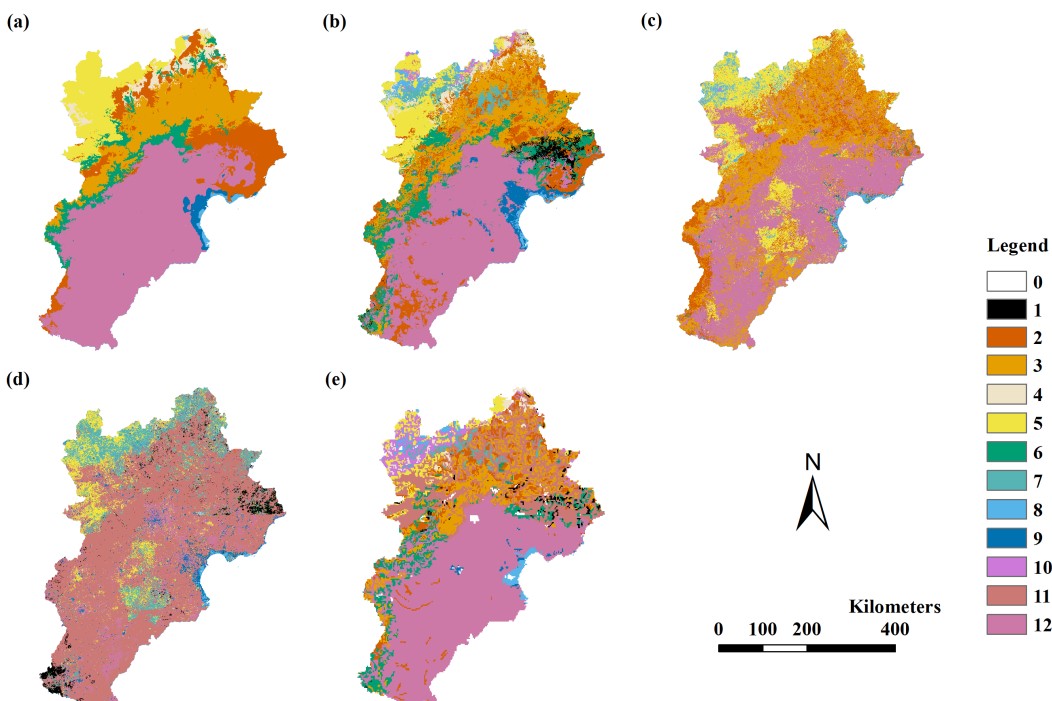

**Figure 3** **The modeling vegetation map of vegetation types with highest accuracy by four methods and the VMC in Jing-Jin-Ji region.** Decision tree model (A), random forest model (B), support vector machine (C), maximum likelihood classification (D), the Vegetation Map of the People's Republic of China (VMC) (E). The legend represents vegetation groups shown in Table 1.

ecosystem diversity, even during global changes (*Faber-Langendoen et al., 2014*). Plants in different vegetation classification units have different spectral characteristics and climatic conditions that are the basis for vegetation distribution simulation. Thus, models using the same variables to simulate the vegetation distribution of different classification units may produce different classification accuracies (*Dobrowski et al., 2008*; *Prasad, Iverson & Liaw, 2006*). Map accuracy has been found to be a function of which classification system and categories are used (*Muchoney et al., 2000*).

Previous studies have explored vegetation distribution simulation using different vegetation classification systems. Plant functional types (PFTs), defined as plant sets sharing similar perturbation response effects on dominant ecosystem processes, have been used to simulate vegetation distribution, as seen in the Biome and Box system models (*Box, 1981*; *Box, 1996*; *Dormann & Woodin, 2002*) with positive simulation results (*Box, 1981*; *Song, Zhou & Ouyang, 2005*; *Weng & Zhou, 2006*). The Mapped Atmosphere-Plant-Soil System (MAPSS) model was also used to simulate vegetation distribution using vegetation life forms, leaf area index, leaf morphology, and leaf longevity (*Zhao et al., 2002*). Other researchers studied potential vegetation distribution using the Holdridge life zone model, with positive vegetation pattern results (*Zheng et al., 2006*). When the IGBP classification system was applied to simulate vegetation distribution at a regional scale, the map estimate accuracy was upwards of 80% (*Muchoney et al., 2000*). In this study, we used machine

**Table 6** **Model assessment of formations and subformations by field point data and VMC.** Variable combinations were shown in Table 3.

| Variable combinations | Decision tree | | | | Random forest | | | | Support vector machine | | | | Maximum likelihood classification | | | |
|---|---|---|---|---|---|---|---|---|---|---|---|---|---|---|---|---|
| | Point data | | VMC | | Point data | | VMC | | Point data | | VMC | | Point data | | VMC | |
| | OA | KC | OA | KC | OA | KC | OA | KC | OA | KC | OA | KC | OA | KC | OA | KC |
| 1 | 23% | 0.14 | 19% | 0.08 | 20% | 0.18 | 5% | 0.02 | 11% | 0.09 | 6% | 0.03 | 8% | 0.06 | 8% | 0.04 |
| 2 | 22% | −0.04 | 49% | 0.04 | 19% | 0.17 | 6% | 0.03 | 13% | 0.11 | 7% | 0.04 | 8% | 0.06 | 13% | 0.05 |
| 3 | 26% | 0.14 | 45% | 0.23 | 29% | 0.27 | 9% | 0.07 | 21% | 0.19 | 10% | 0.07 | 12% | 0.09 | 13% | 0.07 |
| 4 | 30% | 0.20 | 30% | 0.04 | 22% | 0.20 | 7% | 0.04 | 16% | 0.14 | 6% | 0.03 | 9% | 0.07 | 8% | 0.04 |
| 5 | 33% | 0.01 | 67% | 0.00 | 22% | 0.20 | 7% | 0.04 | 15% | 0.13 | 5% | 0.03 | 11% | 0.09 | 10% | 0.04 |
| 6 | 26% | 0.15 | 22% | 0.02 | 31% | 0.30 | 11% | 0.08 | 21% | 0.19 | 8% | 0.06 | 12% | 0.09 | 15% | 0.08 |
| 7 | 33% | 0.20 | 52% | 0.27 | 58% | 0.57[**] | 23% | 0.20 | | | | | | | | |
| 8 | 27% | 0.17 | 34% | 0.18 | 55% | 0.54[*] | 23% | 0.20 | | | | | | | | |
| 9 | 25% | 0.15 | 22% | 0.15 | 55% | 0.53[*] | 22% | 0.20 | | | | | | | | |
| 10 | 30% | 0.17 | 41% | 0.22 | 56% | 0.55[*] | 23% | 0.21 | | | | | | | | |
| 11 | 31% | 0.20 | 41% | 0.22 | 56% | 0.55[*] | 23% | 0.20 | | | | | | | | |

**Notes.**

VMC, the Vegetation Map of the Peoples Republic of China.

[**] the kappa coefficient lager than 0.56.

[*] the kappa coefficient larger than 0.4 and less than 0.56.

OA, Overall accuracy, KC, Kappa coefficient.

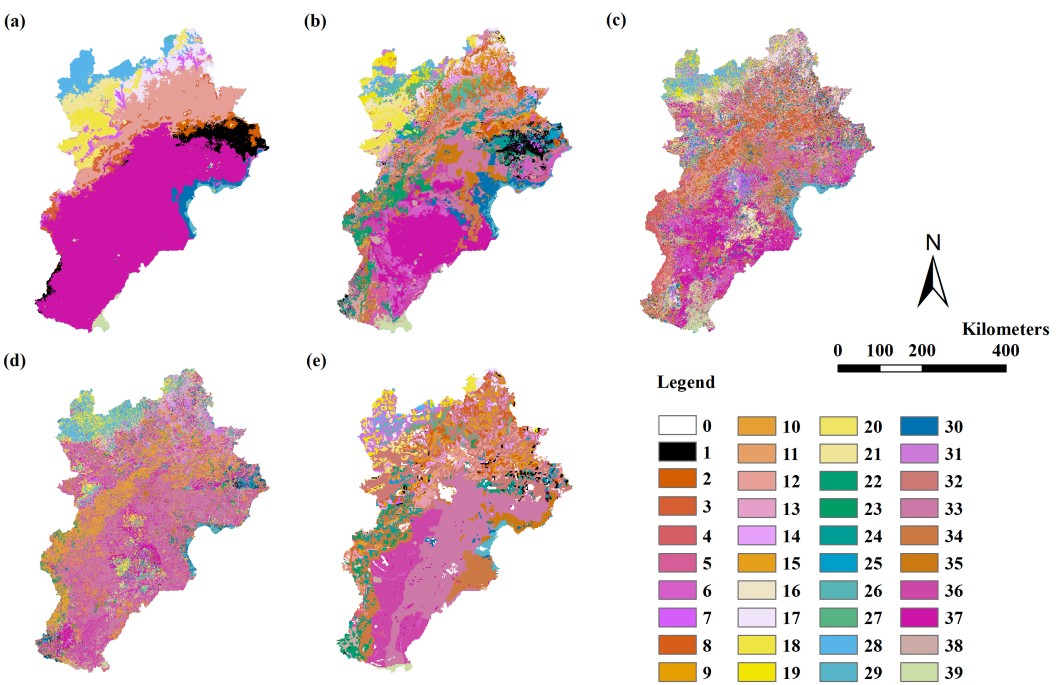

**Figure 4** **The modeling vegetation map of formations and sub-formations with highest accuracy by four methods and the VMC in Jing-Jin-Ji region.** Decision tree model (A), random forest model (B), support vector machine (C), maximum likelihood classification (D), the Vegetation Map of the People's Republic of China (VMC) (E). The legend represents vegetation groups shown in Table 1.

learning models and a hierarchical classification system from the VMC to determine the best modeling method for vegetation affected by high socioeconomic disturbance at various classification levels. In the VMC, unit I was the highest classification level, mainly based upon community appearance; unit II was the second highest level, mainly based upon community and climate appearance; and unit III was the medium classification level, based upon the dominant species. The accuracy of the vegetation distribution simulations in units I and II was similar to each other and higher than unit III's simulation (Tables 4–6).

## Different model performances

We were interested in vegetation distribution modeling's ability to forecast and respond to environmental changes and vegetation pattern management at local to global scales. Vegetation distribution predictions can help explain the relationship between plants and their abiotic and biotic environments (*Franklin, 2010*). To benefit from ecosystem service functions, people can design vegetation distributions according to distribution and abundance patterns and trends (*Hastie, Tibshirani & Friedman, 2009*). Vegetation classification has become a widely used ecological method due to a number of new statistical and machine learning methods used alongside mapped biological and environmental data to model vegetation distributions over large spatial scales at higher resolutions (*Cutler et al., 2007*). Different image classification methods are rarely used together in the same

**Table 7  Top ten most important variables of models in the different vegetation classification units.** The abbreviations of indices were shown in Table 2.

| | Vegetation groups | | | | Vegetation types | | | | Formations and sub-formations | | | |
|---|---|---|---|---|---|---|---|---|---|---|---|---|
| | Decision tree | | Random forest | | Decision tree | | Random forest | | Decision tree | | Random forest | |
| | Important variables | Standardized Importance | Important variables | Normalized importance | Important variables | Standardized Importance | Important variables | Normalized importance | Important variables | Standardized Importance | Important variables | Normalized importance |
| 1 | Annual mean temperature | 1.00 | Annual mean temperature | 3.68 | Annual mean temperature | 1.00 | Annual mean temperature | 3.51 | Annual mean temperature | 1.00 | Annual mean temperature | 4.16 |
| 2 | Annual precipitation | 0.88 | Slope | 2.94 | Slope | 0.83 | Slope | 3.35 | Annual precipitation | 0.86 | Annual precipitation | 3.28 |
| 3 | Slope | 0.80 | Mean diurnal range | 2.60 | Annual precipitation | 0.51 | Mean diurnal range | 3.06 | Slope | 0.63 | Mean diurnal range | 3.25 |
| 4 | Winter vegetation index MR | 0.36 | Annual precipitation | 2.38 | Winter vegetation index MR | 0.30 | Annual precipitation | 2.8 | Mean diurnal range | 0.52 | Slope | 2.24 |
| 5 | Mean diurnal range | 0.33 | Summer vegetation index BI | 1.88 | Mean diurnal range | 0.28 | Summer vegetation index BI | 1.84 | Precipitation of driest month | 0.52 | Precipitation of driest month | 2.16 |
| 6 | Summer surface albedo of band 1 | 0.29 | Winter vegetation index NDVI | 1.37 | Summer vegetation index EVI | 0.22 | Winter vegetation index NDVI | 1.61 | Winter vegetation index MR | 0.4 | Summer vegetation index BI | 1.83 |
| 7 | Summer vegetation index BI | 0.28 | Summer vegetation index EVI | 1.36 | Precipitation of driest month | 0.21 | Winter vegetation index MR | 1.45 | Summer surface albedo of band 1 | 0.32 | Summer vegetation index NDVI | 1.7 |
| 8 | Precipitation of driest month | 0.25 | Winter vegetation index MR | 1.30 | Summer vegetation index BI | 0.20 | Summer vegetation index WI | 1.31 | Summer vegetation index BI | 0.32 | Winter vegetation index NDVI | 1.61 |
| 9 | Summer vegetation index EVI | 0.23 | Summer vegetation index NDVI | 1.22 | Summer surface albedo of band 1 | 0.19 | Precipitation of driest month | 1.24 | Summer vegetation index WI | 0.31 | Winter vegetation index MR | 1.47 |
| 10 | Winter surface albedo of band 6 | 0.19 | Summer vegetation index MR | 1.12 | Winter surface albedo of band 6 | 0.14 | Summer vegetation index NDVI | 1.22 | Winter surface albedo of band 6 | 0.28 | Summer vegetation indices EVI and MR | 1.32 |

classification research, especially when combined with environmental variables (*Li et al., 2014*).

In this study, the RF model performed better than the DT, SVM, and MLC models across the three classification levels. This finding was consistent with the results of other studies that found that the RF method modeled vegetation distribution better than other methods (*Prasad, Iverson & Liaw, 2006*). The DT model divided the data into homogenous subgroups according to the range of predictor variable values. The DT model was generally able to handle a large number of independent variables and could build a tree model faster than the other methods. However, the DT model was somewhat unstable for vegetation distribution modeling and had lower classification accuracy (*Zhou et al., 2016*). The RF model generated a large number of independent trees through data subsets and developed a split in every tree model using a random subset of predictor variables. Therefore, we concluded that the RF model was generally better than the DT model. The SVM model was developed from statistical learning methods and discriminated class samples by locating potentially nonlinear or multiple linear boundaries between individual training points (*Burai et al., 2015*). The aim of the MLC model was to maximize the overall probability that a pixel is correctly assigned to a class. However, the MLC model requires a large number of training samples that limits its application (*Sesnie et al., 2010*). Previous research has shown that classification accuracies when using the SVM classifier were higher than the MLC model (*Pal & Mather, 2005*; *Boyd, Sanchez-Hernandez & Foody, 2006*; *Sanchez-Hernandez, Boyd & Foody, 2007*; *Sesnie et al., 2010*). Because the model had fewer requirements, the DT method provided significantly more accurate classifications than those of the MLC model (*Boyd, Sanchez-Hernandez & Foody, 2006*). Other studies found that the RF and SVM models were similarly accurate (65.3% and 66.6%, respectively) (*Sesnie et al., 2010*), and that the RF, MLC, DT, and SVM models performed similarly and reasonably well when simulating land use classification (*Li et al., 2014*). In addition to the methods mentioned above, an artificial neural network implemented at a regional scale produced classification accuracies of 60%–80% (*Muchoney et al., 2000*; *Haslem et al., 2010*). In the Arctic, this method provided the most accurate vegetation mapping (*Langford et al., 2019*). The reasons for the similarly positive results of these models may be due to the relatively large differences between classification objects, and their use of sufficiently representative training samples and appropriate input variables. In our study, only the SVM and MLC models' output simulated the results of variable Combinations 1 to 6. This may be due to the poor separability of the training samples, as the models could not recognize the training points or their vegetation categories (*Jarnevich et al., 2015*). The Jing-Jin-Ji region has many types of vegetation with very small distribution areas, so the selected training points may have been insufficient. Future training points for these vegetation types should be selected using field surveys, and more suitable models for modeling global vegetation distribution should be developed and tested (*Jiang et al., 2012*).

## Important variables in vegetation classification models

Variable selection is directly related to the vegetation distribution model's ability to capture important environmental factors (*Mod et al., 2016*). Models predict the important

variables that drive the distribution of vegetation (*Prasad, Iverson & Liaw, 2006*). Vegetation distribution is predominantly driven by temperature, precipitation, and topographical variables (*Franklin, 1995*; *Mod et al., 2016*; *Prasad, Iverson & Liaw, 2006*), specifically those related to physiological tolerance, site energy, and moisture balance (*Franklin, 1995*). In addition to environmental variables, some spectral variables are used as input variables. However, the overuse of spectral variables can actually decrease discrimination accuracy, meaning that only spectral variables reflecting vegetation information should be selected, such as those related to the visible spectrum, infrared spectrum, and vegetation indices (*Price, Guo & Stiles, 2002*; *Zhou et al., 2016*). Different variables respond to different information. Spectral variables directly reflect land surface object information, while geospatial and climatic variables reveal information about the vegetative environment.

Terrain, an important variable in vegetation distribution models, has long been used to improve map accuracy, especially for regions with large elevation differences (*Dobrowski et al., 2008*; *Oke & Thompson, 2015*). *Sesnie et al. (2010)* found that adding elevation as a predictive variable dramatically improved the accuracies of the SVM and RF models >80% for most forest types. Slopes with similar elevations but different aspects have very different soil and vegetation temperatures (*Gunton, Polce & Kunin, 2015*; *Mod et al., 2016*). *Dobrowski et al. (2008)* highlighted the importance of slope and aspect when mapping vegetation communities in the Sierra Nevada. Slope was also an important variable in this study (Table 7) since different types of vegetation require different precipitation and temperature levels and have different tolerances to extreme heat and cold. The significance of these climate variables (annual mean temperature, temperature range, and annual precipitation) has been validated in other studies (*Prasad, Iverson & Liaw, 2006*; *Sesnie et al., 2008*). We looked at two surface albedo indices (the summer surface albedo of band 1 and the winter surface albedo of band 6). *Sesnie et al. (2010)* combined elevation and spectral band data to increase the classification accuracy to a satisfactory level for most forest types. *De Colstoun et al. (2003)* obtained high accuracies (80%) when classifying coniferous, temperate broad-leaf, and mixed forest types using Landsat ETM+ bands. Other studies have used different vegetation index variables (*Price, Guo & Stiles, 2002*; *Zhou et al., 2016*) specific to their study areas and data.

The input variables used in our vegetation distribution model are not exhaustive. Ecophysiologically meaningful predictors such as soil moisture, pH, and nutrients, should be considered. Other factors, such as actual light, disturbance, biotic interactions, land use, and bioclimatic information could also be incorporated into vegetation distribution models (*Dobrowski et al., 2008*; *Mod et al., 2016*; *Prasad, Iverson & Liaw, 2006*; *Sesnie et al., 2010*). We suggest building more ecophysiologically sound vegetation distribution models that require a collaborative effort across the ecological, geographical, and environmental sciences (*Mod et al., 2016*).

## Other factors affecting classification accuracy

In addition to classification units, models, and input variables, classification accuracy is affected by other factors, including algorithm error and image data (*Li et al., 2014*). We must acknowledge the existence of errors in random sample selection, modeling, and data

preprocessing algorithms. Remote sensing data sources, as well as the date and processing of selected images, vary, resulting in different simulated values and accuracies (*Price, Guo & Stiles, 2002*). Remote sensing images with high spectral and spatial resolutions provide rich spectral and ground information, moderately improving the predictive ability of the vegetation distribution model (*Peng et al., 2002*). However, the use of high spectral and spatial resolution images creates a greater demand for data access, larger computer storage capacities, and faster data processors (*Price, Guo & Stiles, 2002*), which is why we did not use high spectral and spatial resolution images in this study. Moreover, some cultivated vegetation and shelter forests in the Jing-Jin-Ji region are greatly affected by human disturbance, which affects their water-heat conditions and soil nutrition. Urbanization reduces vegetation, transforming some areas into industrial, commercial, and residential land. This has led to the direct or indirect pollution of the water, soil, and air, and the reduced predictive ability of vegetation distribution models. The VMC we used for model assessment was published in 2007, and no updated study has been published over the past 10 years. The current state of the Jing-Jin-Ji region's vegetation no longer coincides with the VMC's assessment.

## CONCLUSIONS

Our main objective was to determine the best simulation method for vegetation affected by high socioeconomic disturbance in the Jing-Jin-Ji region. The RF model was the most capable at simulating vegetation distribution across all three units. The DT model could simulate the vegetation distribution in units I and II. The SVM and MLC models could not simulate the distribution in any of the three units. Based on the Kappa coefficient, the RF model was generally better than the DT model and the most suitable model for simulating vegetation distribution in the Jing-Jin-Ji region. The most important variables affecting vegetation classification accuracy were three climate variables (annual mean temperature, mean diurnal range, and annual precipitation), one geospatial variable (slope), and two spectral variables (Mid-infrared ratio of winter vegetation index and brightness index of summer vegetation index). We recommend using the RF model to produce or improve the vegetation maps in areas of high human disturbance.

## ACKNOWLEDGEMENTS

We thank two anonymous reviewers and the editor for their effort to review this manuscript.

### Funding

This work was funded by the National Key R & D Program of China (2018YFC0506903). The funders had no role in study design, data collection and analysis, decision to publish, or preparation of the manuscript.

## Grant Disclosures

The following grant information was disclosed by the authors:
National Key R & D Program of China: 2018YFC0506903.

## Competing Interests

The authors declare there are no competing interests.

## Author Contributions

- Sangui Yi conceived and designed the experiments, performed the experiments, analyzed the data, prepared figures and/or tables, authored or reviewed drafts of the paper, and approved the final draft.
- Jihua Zhou conceived and designed the experiments, prepared figures and/or tables, and approved the final draft.
- Liming Lai performed the experiments, prepared figures and/or tables, and approved the final draft.
- Hui Du performed the experiments, authored or reviewed drafts of the paper, and approved the final draft.
- Qinglin Sun analyzed the data, authored or reviewed drafts of the paper, and approved the final draft.
- Liu Yang and Xin Liu analyzed the data, prepared figures and/or tables, and approved the final draft.
- Benben Liu analyzed the data, prepared figures and/or tables, authored or reviewed drafts of the paper, and approved the final draft.
- Yuanrun Zheng conceived and designed the experiments, prepared figures and/or tables, authored or reviewed drafts of the paper, and approved the final draft.

## Data Availability

Data is available at the 4TU.Centre for Research Data: Yi, Sangui (2020): Sample data for simulation in Jing-Jin-Ji region. 4TU.ResearchData. Dataset. https://doi.org/10.4121/uuid:1b27dc6b-b77e-4f18-b035-e8a249f595c0.

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
