# Peer review of "Simulating highly disturbed vegetation distribution: the case of China’s Jing-Jin-Ji region"

_PeerJ, doi:10.7717/peerj.9839_

## Round 0.1 · original submission · Major Revisions

I have now received feedback from two qualified reviewers regarding your manuscript "Simulation of the vegetation distribution in the Jing-Jin-Ji region that has been highly disturbed by social-economic development".

Though both reviewers acknowledge that the study is potentially valuable, they raise substantial concerns that should be thoroughly addressed before the paper can be further considered for publication. Both note the need for a substantial proofreading of the paper. More importantly, they both raise a number of issues regarding the methodology of the study (see especially section "Experimental design" for reviewer #1 and the "Comments for the author" section for reviewer #2): I think these are important points that should all be clearly and adequately addressed in the revised version, should you decide to submit one.

I hope the reviews will prove to be valuable to you in improving the quality of the manuscript.

Reviewer 1 ·

Basic reporting

The reviewed manuscript is comprehended only with great difficulty. The text is on many occasions ambiguous and often lacking strong arguments. The text needs major improvements in English language, consistency and clarification. Both the introduction and the discussion are lacking a coherent structure.
Long difficult-to-understand sentences: line 48-52; line 213-217; lines 321-328
Comma problems: line73 …
vague language: line 78-86; line 189, line 238-239
out of context: line 98-99,
rephrasing required: line 59 “smartly use vegetation”; line 69 “using information based on information..”; line 189 “through different model methods, we used different combinations…”; line 299 “a complex, multi-level, non-linear system, is one of the complex problems…;
repetition of methods: line 359-362
language problems: line 122; line 345



Manuscript uses broad and sufficiently varied literature, however it often relies on literature from a single book (Franklin 2010) without mentioning page numbers within this book (all instances when the book is cited) and often lacking reasoning (line 359).
Statement unsupported by reference: line 64; line 354
statement without reference: line 102-103

@Franklin, J. (2010). Mapping species distributions: spatial inference and prediction, Cambridge University Press.



Figures are generally in low resolution (and no hi-resolution/original resolution GeoTiff is provided). Color schemes used are difficult to distinguish and not color-blind friendly (example in Figure 4, where 39 different colors are used – impossible for human eye to distinguish).
Figure captions are cut-off (Figure 2,3,4).
Figure 1 is missing an overview-localization map (or marked large cities), for the readers who are not familiar with geography of China.

Raw data is partially shared, but point files are in uncommon format that is difficult to open using common computer programs (and not specified which program should be used). Additionally, the provided shapefile data for Vegetation Map (VMC) contains descriptions and encoding in non-latin and non-English characters, thus could not be explored further.

Experimental design

This research is within the scopes of PeerJ journal, however it seems very repetitive and little original with regard to the previous article from the authors Zhou et al. from the year 2016. There is no significant new findings or methodological advancements (default settings of an existing software) other than the produced vegetation map predictions (which should at least be provided in full resolution)
@Zhou, J., et al. (2016). "Comparison modeling for alpine vegetation distribution in an arid area." Environmental monitoring and assessment 188(7): 408.

The main flaw is in the fact that the authors are using relatively old vegetation survey data (that they claim is from year 1980 – line 121), which is then put through the modelling procedures using recent predictor variables (including vegetation indices from 2013). This could be causing a problematic temporal mismatch, especially since the study area is reported to be are heavily influenced by socio-economic development.


The software that was used to carry out the entire study is very little described (EnMAP-Box), and only mentioned at line 220. Describe the modelling platform early in this paragraph (line 202). The default settings are used, but it would be useful to elaborate further on what are the settings. Why was it left on default? Other studies that only use the default settings?

Lacking description of how data between training and assessment points were split (Random subset?).

Not described how the vector vegetation map of China (VMC) was used as assessment. I must only assume that the vector map was first rasterized onto the same grid as the modelled data, and then tested with Kappa. Please explain further

Kappa was used as the only assessment method. It is known that Kappa is a threshold dependent metric, see Guisan 2017 page 242 for additional assessment methods

@Guisan, A., et al. (2017). Habitat suitability and distribution models: with applications in R, Cambridge University Press.

Validity of the findings

It is common in the Distribution Modelling (DM) research to provide all input data on a repository for possibility of future replication. It is highly advisable to do so in this study. See Araújo et al. (2019) for standards for DM
@Araújo, M. B., et al. (2019). "Standards for distribution models in biodiversity assessments." Science Advances 5(1): 1-11.

Conclusion paragraph both in the abstract and discussion is a word-by-word repetition of the results.
Line 329 – there is no explanation to support your statement.

Additional comments

I appreciate the last few sentences in discussion (line447-451) taking up the problem of human disturbance to the study area. This should be elaborated on in much greater extent.

The title should be reconsidered: “region that has been highly disturbed by social-economic development” it is not the core topic of this study, and the authors have avoided explaining/discussing or performing analysis on the human disturbances – rather they are modelling the potential vegetation

Shortcuts are not used consistently throughout the article (DT, RF, SVM and MLC sometimes written out, sometimes used with shortcut)

Use table to present the variable combinations (starting on line 187). Variable combination 11 and 12 are not sufficiently explained (give explicit examples – create a overview table).

How can predictor variables be used for assessment? Line 189

When listing out the three classification units, try to delimit the units with numbers (I. vegetation groups; II. vegetation types; and III. formations and subformations) to avoid double “and”

·

Basic reporting

I see many text errors thoughout the manuscript and some of the statements can definitely be written in a better way. I suggest that the co-authors whose first language is English carefully peruse the text. Most sentences are long and superfluous. Here are some of the few examples:
1. Line 19 - Vegetation distribution simulations could help to understand..may be better to write as: Vegetation distribution simulation is important to understand ….
2: lines 48 - 52: Very long sentence and difficult to understand. It can benefit from paraphrasing and breaking into two or three sentences.
3. Caption of figure 3 – “label e” is lacking … the Vegetation Map of the People's Republic of China in (e)
4. Similarly, on caption of figure 4, “label e” is lacking
5. Please supplement the 3763 observation points with corresponding vegetation groups, types, and formations and sub-formations.

Experimental design

The experimental design is good. But I have some concerns on resampling techniques and variable selections. Below I am also suggesting an alternative way of achieving the same objective.

Validity of the findings

The findings are valid but the objective is not clear. The paper will benefit by setting a clear objective - whther the primary goal of the paper is to improve the vegattion map of the study area or to explore and answer methodological questions. This should be clearly mentioned in the results and conlcusion sections too.

Additional comments

1. The resampling techniques used: nearest-neighbor method is used to resample both the DEM and the climatic variables. This approach is commonly used for discrete variables such as land-cover. For continuous variables, bilinear or cubic techniques are commonly used. The authors should justify either how their approach may not affect their data and consequently their results or should use the right approach.
2. Variable selections: what is the criteria to select the four bioclimatic variables? Is there any objective way to determine the number of bioclimatic variables to be used as predictor variables? It is better to consider all the 19 bioclimatic variables, stack them together with the other predictor variables, extract their values at the 3763 observation point, compute pair wise correlation and consider only less correlated variables (R < /0.7/ is usually used as threshold; see Chala et al., 2017). This way, it is possible to be objective and avoid redundantly using correlated variables. It is also interesting to see the correlation among the topographic, spectral and climate variables.
3. Elevation is a proxy variable and does not have direct impact on vegetation. Thus I don’t recommend using elevation as a predictor variable. It is also expected to be highly correlated with temperature related bioclimatic variables. Either the use of altitude as predictor variable should be justified or its correlation with the bioclimatic variables should be checked.
4. Probablity of using alternative approaches: cultural vegetation groups and the vegetation types included in this category are mainly shaped by anthropogenic impacts. Including this group in the model will definitely deteriorate model quality. I am very curious what if you just use satellite images with higher spatial resolution such as sentinel and spot images and check how they perform in capturing the vegetation groups, types, formations and sub formations. That asks less energy and probably more reliable. That will also allow detecting anthropogenic driven changes and monitoring the land-use land-cover of the study area. Specially of the main objective of the paper is to explore the method that perfoms well, this vegetation types should be excluded from the model.
5. Checking the homogeneity of classes: It is also good to check the elevation ranges of each vegetation groups and types. If they cover wider elevation ranges, by dividing them into upper and lower elevation classes, it is possible to improve the performances of the algorithms (See Chala et al., 2017).
6. Figure 1 – it is great that the elevation range of the study area is provided. But it will be more informative if it is classified in to at least five reasonable elevation range classes. Can you please consider that?
7. The title can be kept short and precise – for example: Simulation of highly disturbed vegetation types: the case of Jing-Jin-Ji region, China
8. Last and finally: Is it possible to include the specific objective of the paper in the abstract – whether it is more interested in improving the vegetation map of the study area or exploring methodological issues. Just a sentence or two?
Reference: Chala, D., et al. (2017). "Migration corridors for alpine plants among the ‘sky islands’ of eastern Africa: do they, or did they exist?" Alpine Botany 127(2): 133-144.

---

## Round 0.2 · Minor Revisions

Dear Dr. Yi,

having carefully examined the revised manuscript myself, I agree with the reviewer's assessment that the submission has substantially improved with respect to the original version.

Reviewer #2 raises some additional points that I kindly ask you to address in a revised version.

In addition, I kindly ask you to consider also the following points when revising and resubmitting your manuscript. Main comments:

1. the novelty of the work could be better highlighted in the paper. I make here direct reference to comment 6 by reviewer #1 in the original submission. I am satisfied with the answer you give to this comment in the rebuttal letter, but this should be included in the manuscript as well. In particular, you should acknowledge the link with Zhou et al (2016), while explaining that the present work is sufficiently distinct. I think the end of the Introduction (L105) could be a good place for this.

2. with reference to comment 8 by reviewer #1 in the original submission, while you have satisfactorily improved the description of methods in the revised submission, I think the role of the "EnMAP-Box software" still remains rather obscure and should be better clarified.

Minor comments:
3. L19: a word seems to be missing here after "improving the vegetation..."

4. L50: "Industrialization" is not the only source of alterations in vegetation patterns (e.g., urbanization, population growth, land use change for agricultural use, etc.). Please revise to acknowledge other pressures that are relevant in the study region in addition to industrialization.

5. L88: I don't think "magnitude" is the correct word here. Maybe "extension"?

6. L93: remove "more" before "limited resources"

7. L130: please add a sentence to clarify what were the criteria for selecting 8, 12 and 39 units from the three different groups.

8. L141: a reference is needed for the "Kappa coefficient".

9. L168: please clarify what type of correlation the "R" presented here refers to (e.g., Pearson, Spearman, etc.)

10. L357: I think the title of this revised section could be more descriptive. Perhaps "Important variables in vegetation classification models"?

11. L420: please add "in the Jing-Jin-Ji region" at the end of the sentence currently ending with "socioeconomic disturbance".

·

Basic reporting

The article is much improved now except there are still some minor editorial mistakes.

Experimental design

This part is already improved

Validity of the findings

It is quite improved and good

Additional comments

The authors included (justified) most of the comments which were given on the earlier version. But stil there are few minor editorial mistakes. They are presented as follows:
18 - May be it can improved as: The primary goal of this study was to determine the best simulation method for a vegetation in an area that is heavily affected by human disturbance.
35 - "Variables Combination 7 produced the highest simulation accuracy". I think it is better to list the variables by name.
63 - "in conjunction": may be better to use in combination?
145 - "digital elevation model (DEM) (Zhao et al., 2018)" - may better to write it as - digital elevation model (DEM; Zhao et al., 2018).
148 - "1 km resolution" - may be better to put it as - at ~ 1 km resolution? I don't think the resolution is exactly 1 km. Please cross check this.

---

## Round 0.3 · accepted · Accept

All the comments on the previous version were satisfactorily addressed. The current version of the manuscript can be accepted for publication. Congratulations to the authors!